# Orchard layout and plant traits influence fruit yield more strongly than pollinator behaviour and density in a dioecious crop

Angela Peace[1]*, David Pattemore[2,3], Melissa Broussard[2], Dilini Fonseka[1], Nathan Tomer[2], Nilsa A. Bosque-Pérez[4], David Crowder[5], Allison K. Shaw[6], Linley Jesson[7], Brad G. Howlett[8], Mateusz Jochym[2], Jing Li[9]

**1** Department of Mathematics and Statistics, Texas Tech University, Lubbock, TX, United States of America, **2** The New Zealand Institute for Plant and Food Research, Hamilton, New Zealand, **3** School of Biological Sciences, University of Auckland, Auckland, New Zealand, **4** Department of Entomology, Plant Pathology and Nematology, University of Idaho, Moscow, ID, United States of America, **5** Department of Entomology, Washington State University, Pullman, WA, United States of America, **6** Department of Ecology, Evolution and Behavior, University of Minnesota, St. Paul, MN, United States of America, **7** The New Zealand Institute for Plant and Food Research, Havelock North, New Zealand, **8** The New Zealand Institute for Plant and Food Research, Lincoln, New Zealand, **9** Department of Mathematics, California State University Northridge, Northridge, CA, United States of America

* a.peace@ttu.edu

**Data Availability Statement:** All relevant data are within the manuscript and its Supporting Information files.

## Abstract

Mutualistic plant-pollinator interactions are critical for the functioning of both non-managed and agricultural systems. Mathematical models of plant-pollinator interactions can help understand key determinants in pollination success. However, most previous models have not addressed pollinator behavior and plant biology combined. Information generated from such a model can inform optimal design of crop orchards and effective utilization of managed pollinators like western honey bees (*Apis mellifera*), and help generate hypotheses about the effects of management practices and cultivar selection. We expect that the number of honey bees per flower and male to female flower ratio will influence fruit yield. To test the relative importance of these effects, both singly and simultaneously, we utilized a delay differential equation model combined with Latin hypercube sampling for sensitivity analysis. Empirical data obtained from historical records and collected in kiwifruit (*Actinidia chinensis*) orchards in New Zealand were used to parameterize the model. We found that, at realistic bee densities, the optimal orchard had 65-75% female flowers, and the most benefit was gained from the first 6-8 bees/1000 flowers, with diminishing returns thereafter. While bee density significantly impacted fruit production, plant-based parameters-flower density and male:female flower ratio-were the most influential. The predictive model provides strategies for improving crop management, such as choosing cultivars which have their peak bloom on the same day, increasing the number of flowers with approximately 70% female flowers in the orchard, and placing enough hives to maintain more than 6 bees per 1000 flowers to optimize yield.

**Funding:** This work was supported by Plant & Food Research Discovery Science grant DS 14-65. The funders had no role in study design, data collection and analysis, decision to publish, or preparation of the manuscript.

**Competing interests:** The authors have declared that no competing interests exist.

## Introduction

Mutualistic plant-pollinator interactions play a vital role in plant reproduction in both natural systems and managed (i.e. agricultural) systems. Animal-mediated pollination is important for 87.5% of angiosperms globally [1], and 75% of the most important crop species benefit significantly from this service [2], providing greater than US$170 billion in economic value annually [3]. Functionally dioecious plants are especially reliant on pollination, as pollinators must cross from one plant to another. Even in well-studied systems, such as kiwifruit (*Actinidia chinensis*), the complexity of interacting variables limits the ability of researchers to provide clear recommendations to growers, with proposed stocking rates varying from 3-8 colonies per ha [4].

Mathematical modeling of plant-pollinator interactions can help understand key determinants in pollination success [5]. Such approaches could be valuable tools for designing optimal crop orchard layouts and for the effective use of managed pollinators in agricultural systems. This may be especially important in dioecious crops that have separate male and female plants which adds further complexity in conducting empirical field trials when these plants respond differently to environmental variables.

In spite of this, pollination models have tended to focus on plant biology [6–9] or insect behavior [10, 11] but few have examined both simultaneously [12, 13]. Including variables such as flower phenology, the ratio of male to female flowers, pollinator abundance, and flower handling behavior could assist in the generation of robust models. Combining information from both pollinators and plants in the same framework more realistically represents field conditions and enables us to directly compare their importance. A significant challenge in developing good models is sufficient data for parameterization.

We chose kiwifruit as our model dioecious crop system as there are four decades of empirical data, examining many aspects of both insect behavior and plant biology [14]. Kiwifruit is a deciduous vine, with male and female flowers borne on separate plants [15]. Neither sex produces nectar, and the female flowers produce inviable pollen [15], which is high in lipids [16], to attract pollinators instead. Plants are typically trained onto a pergola system, with male vines interplanted amongst a larger number of female vines at a 1:3 to 1:8 ratio [4, 17]. Although male cultivars typically have 2-3x more flowers than female cultivars [18], grower planting and pruning regimes ultimately determine the floral sex ratio in orchards. While kiwifruit have a number of pollinating species in their native range [19–21], most growing regions rely on honey bees for pollination, representing the vast majority of all flower visitors in the United States [22], France [23], Australia [24], and New Zealand [25, 26].

We expected that male-female kiwifruit flower ratio and pollinator density will influence fruit yield, along with various parameters of pollinator behavior. To test the relative importance of these effects, both singly and simultaneously, we used a system of delay differential equations (DDEs) combined with Latin hypercube sampling for parameter sensitivity analysis [27]. The model explicitly tracks pollinators (parameterized here based on data from honey bees), with varying pollen loads as they preferentially visit male and female flowers, as well as the current number of open flowers over time. The delays incorporated into this model take into account the lifespan of open flowers, as male and female flowers open and close throughout the blooming period.

## Materials and methods

### Model development

We develop and analyze a mathematical model of pollination dynamics that incorporates key aspects of both plant biology and insect behaviors. First, we present a sub-model of the

flowering dynamics in an orchard, then in the following section, we add the pollinator dynamics. We assume homogeneous conditions across the field for both flower and pollinator densities. Table 1 describes the model state variables and parameters.

**Flowering dynamics.** We consider a kiwifruit orchard made up of male and female trees and model the opening and closing of flowers throughout the bloom. To capture pollination dynamics, it is important to know how many male and female flowers are open at any given day. Here, we assume that the total number of flower buds in the field is fixed and the rate they open follows a normal distribution. Let $B_m$ and $B_f$ denote the total number of male and female flower buds. Initially all flower buds are closed. Let $M$ and $F$ denote the number of male and female flowers that have opened. The rates that these flowers open is modeled as

$$\frac{dM}{dt} = \frac{B_m}{\sqrt{2\pi\sigma_m^2}} e^{-\frac{(t-t_m)^2}{2\sigma_m^2}} \tag{1a}$$

$$\frac{dF}{dt} = \frac{B_f}{\sqrt{2\pi\sigma_f^2}} e^{-\frac{(t-t_f)^2}{2\sigma_f^2}} \tag{1b}$$

**Table 1. Description of model state variables and parameters.**

| Flower dynamics | |
|---|---|
| **Variable** | **Meaning** |
| $m(t)$ | number of open male flowers at time $t$ |
| $f(t)$ | number of open female flowers at time $t$ |
| **Parameter** | **Meaning** |
| $B_m$ | number of male buds |
| $B_f$ | number of female buds |
| $t_m$ | peak day of male flower opening rate |
| $t_f$ | peak day of female flower opening rate |
| $\sigma_m$ | spread of male flowering period |
| $\sigma_f$ | spread of female flower period |
| $\tau_m$ | life span of male flowering |
| $\tau_f$ | life span of female flowering |
| **Pollinator dynamics** | |
| **Variable** | **Meaning** |
| $P_{m1}$ | pollinators with high pollen loads |
| $P_{m2}$ | pollinators with medium pollen loads |
| $P_{m3}$ | pollinators with low pollen loads |
| $P_f$ | pollinators carrying no pollen |
| **Parameter** | **Meaning** |
| $\rho$ | pollinators per 1000 flowers |
| $\alpha$ | search rate |
| $\beta$ | handling time |
| $\delta$ | preference to remain on male flowers |
| $\varepsilon$ | preference to remain on female flowers |
| $p_1$ | percent chance to set fruit from single type one visit |
| $p_2$ | percent chance to set fruit from single type two visit |
| $p_3$ | percent chance to set fruit from single type three visit |

where $t_m$ and $t_f$ are the times when the opening rates are highest and $\sigma_m$ and $\sigma_f$ are the variations in these rates of opening. We assume that each flower is only open for a fixed amount of time. Male flowers are open for $\tau_m$ days and female flowers remain open for $\tau_f$ days. While $M$ and $F$ denote the total number of male and female flowers that have opened, the number of currently opened flowers changes, as flowers close. Let $m$ and $f$ denote the number of male and female flowers that are currently open, which can be determined with the following expressions:

$$m(t) = \begin{cases} M(t) - M(t - \tau_m) & \text{if } t > \tau_m \\ M(t) & \text{elsewhere} \end{cases} \quad (2a)$$

$$f(t) = \begin{cases} F(t) - F(t - \tau_f) & \text{if } t > \tau_f \\ F(t) & \text{elsewhere.} \end{cases} \quad (2b)$$

Example simulated dynamics of open flowers over time are depicted in Fig 1. Here, we chose and varied some parameter's values to highlight the role they have in shaping the curve describing flowering dynamics.

**Pollinator dynamics.** Pollinator dynamics are modeled with differential equations that divide the population into subcompartments based on their pollen load. Pollinators can have a high, medium, or low pollen load (denoted as $P_{m1}$, $P_{m2}$, and $P_{m3}$ respectively) or be carrying no pollen (denoted at $P_f$). These states represent a division of empirical data on single-visit deposition, which often follows an exponential [28] or steeper than exponential decay [29].

We assume that pollinators completely load up on pollen with a visit to a male flower and deposit some pollen with a visit to a female flower. We assume that male pollen availability is not limiting in this scenario; to partially compensate for this short-coming of the model we limit active foraging to four hours per day as captured by the visitation rate, corresponding to field observations [30]. This four-hour window of pollen-foraging activity limits the total amount of pollination in a day (built into the visitation rate parameter). Within this window, pollen availability is typically not a limiting factor in kiwifruit due to male flowers having up to twice as many pollen grains than female flowers, and the anthers continued to dehisce over this four-hour period. A diagram depicting the movement of pollinators between the compartments is shown in Fig 2. A pollinator with a high pollen load can move between compartments $P_{m1}$, $P_{m2}$, $P_{m3}$, and $P_f$ with subsequence visits to female flowers. Regardless of current pollen loads, whenever a pollinator visits a male flower it completely loads up on pollen and enters the $P_{m1}$ compartment.

The rate that pollinators visit male and female flowers is a crucial part of the model dynamics. We consider a pollinator visitation rate that depends on the search rate ($\alpha$), the handling time ($\beta$) and the densities of open male ($m$) and female ($f$) flowers, as described above in Eq (2). For pollinator visitation rates, previous work suggests that saturating functions of flower densities such as Holling type II functional responses are typical of oligolectic consumers that use only a few plant species and Holling type III responses are typical of generalist consumers that switch between hosts [6]. While honeybees are generalist, here we use a Holling type II response because of the mono-culture orchard environment in the model. Following previous studies [6, 31, 32] we defined the total pollinator visitation rate as:

$$\text{Total pollinator visitation rate} = \frac{\alpha(f + m)}{1 + \alpha\beta(f + m)} \quad (3)$$

which has the units of per time. This visitation rate includes visits to both male and female

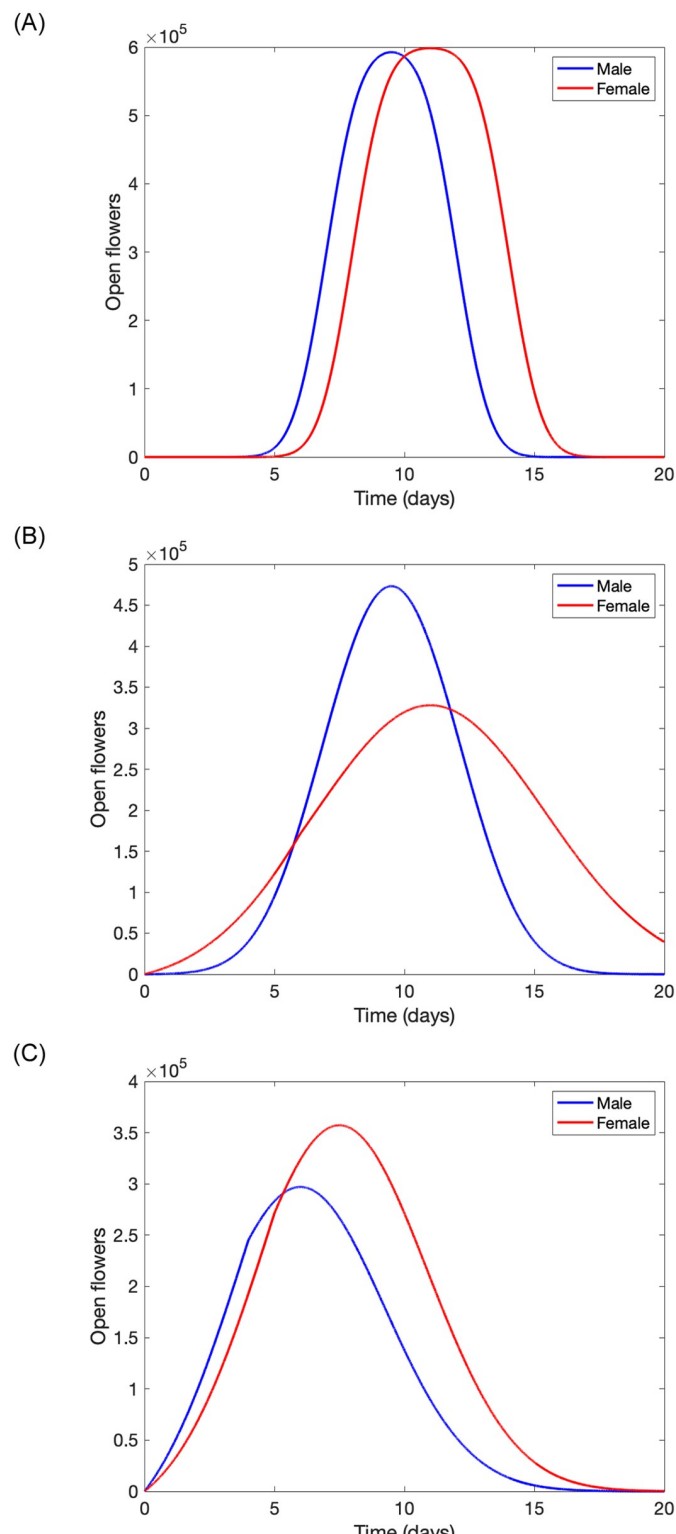

**Fig 1. Example simulations of open flowers over time following Eq (2) starting with 600,000 of each male and female flower buds $B_m = B_f = 600,000$ (representing a one ha. of the orchard) for (a) $\sigma_m = 1, \sigma_f = 1, t_m = 7, t_f = 8, \tau_m = 5, \tau_f = 6$ and (b) $\sigma_m = 2, \sigma_f = 4, t_m = 7, t_f = 8, \tau_m = 5, \tau_f = 6$ and (c) $\sigma_m = 3, \sigma_f = 3, t_m = 4, t_f = 5, \tau_m = 4, \tau_f = 5$.**

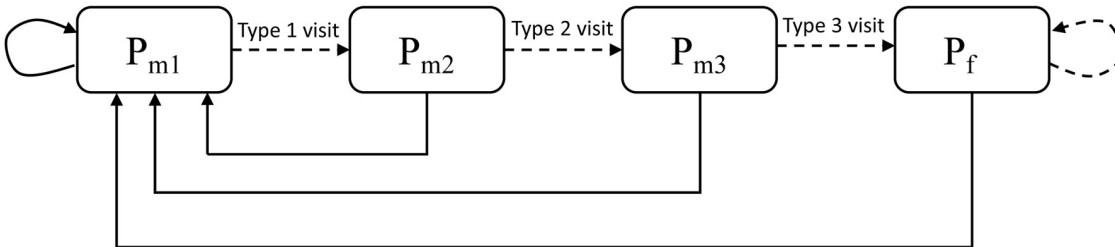

**Fig 2. Model flow diagram of pollinator types.** Solid lines depict visits to a male flower. Dashed lines depict visits to a female flower.

flowers. The movement of pollinators between male and female flowers depends on the proportion of male vs female flowers, as well as pollinator preferences. Previous studies suggest that honey bees have a preference to visits flowers of the same sex as the one they are currently on [22, 24, 33, 34]. We define the preference parameter $\delta$ such that a pollinator on a male flower can preferentially choose to next visit another male flower ($0 < \delta < 1$). Similarly, we define the preference parameter $\epsilon$ such that a pollinator on a female flower preferentially next visits another female flower ($0 < \epsilon < 1$). Pollinators have no preference if $\delta = \epsilon = 1$. We used these preference parameters to define functional forms representing the probability of a pollinator to visit each type of flower, following the method used in [35]. These probabilities depend on the ratio of male to female flowers raised to the power of the preference, such that the movement of pollinators between flowers can be written as the following expressions:

$$\left(\frac{m}{f+m}\right)^{\delta} \qquad \text{fraction on male flowers that move to a male flower} \qquad (4a)$$

$$1 - \left(\frac{m}{f+m}\right)^{\delta} \qquad \text{fraction on male flowers that move to a female flower} \qquad (4b)$$

$$\left(\frac{f}{f+m}\right)^{\varepsilon} \qquad \text{fraction on female flowers that move to a female flower} \qquad (4c)$$

$$1 - \left(\frac{f}{f+m}\right)^{\varepsilon} \qquad \text{fraction on female flowers that move to a male flower} \qquad (4d)$$

Fig 3 shows plots of the probabilities of pollinators that will switch the type of flower they are on verses the proportion of female flowers in the orchard, for various preferences parameters.

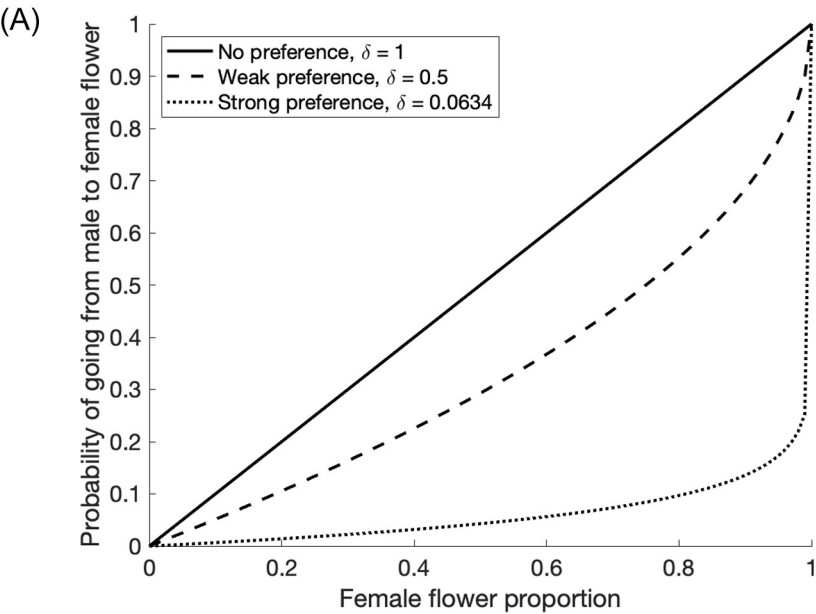

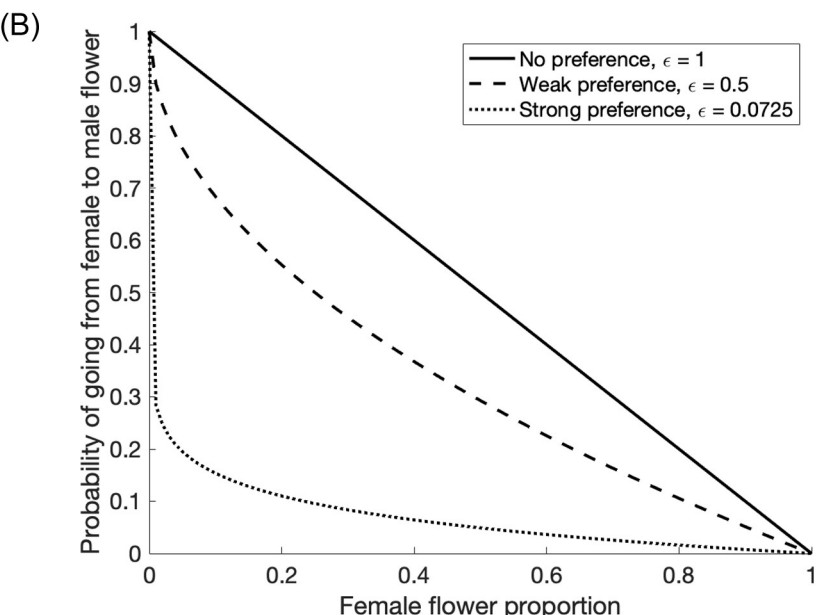

**Fig 3. Movement probabilities for (a) pollinators on male flowers switching to a female flower and (b) pollinators on female flowers switching to a male flower for strong, weak, and no preferences in paramters $\delta$ and $\epsilon$.** Values for strong preferences were used for the analyses in this paper, details are described in the Parameterization section. Note that a strong preference for remaining on the same type of flower corresponds with a low probability of switching between flower types.

**Full pollinator-flower model.** The complete pollinator-flower model are described with the following system of differential equations:

$$\frac{dP_{m1}}{dt} = \underbrace{\left(\frac{\alpha(f+m)}{1+\alpha\beta(f+m)}\right)}_{\text{Total visitation rate}} \left[ \underbrace{\left(1-\left(\frac{f}{f+m}\right)^{\varepsilon}\right)(P_{m2}+P_{m3}+P_f)}_{\text{moves from female to male}} - \underbrace{\left(1-\left(\frac{m}{f+m}\right)^{\delta}\right)P_{m1}}_{\text{moves from male to female}} \right] \quad (5a)$$

$$\frac{dP_{m2}}{dt} = \underbrace{\left(\frac{\alpha(f+m)}{1+\alpha\beta(f+m)}\right)}_{\text{Total visitation rate}} \left[ \underbrace{\left(1-\left(\frac{m}{f+m}\right)^{\delta}\right)P_{m1}}_{\text{moves from male to female}} - \underbrace{\left(1-\left(\frac{f}{f+m}\right)^{\varepsilon}\right)P_{m2}}_{\text{moves from female to male}} - \underbrace{\left(\frac{f}{f+m}\right)^{\varepsilon}P_{m2}}_{\text{moves from female to female}} \right] \quad (5b)$$

$$\frac{dP_{m3}}{dt} = \underbrace{\left(\frac{\alpha(f+m)}{1+\alpha\beta(f+m)}\right)}_{\text{Total visitation rate}} \left[ \underbrace{\left(\frac{f}{f+m}\right)^{\varepsilon}P_{m2}}_{\text{moves from female to female}} - \underbrace{\left(1-\left(\frac{f}{f+m}\right)^{\varepsilon}\right)P_{m3}}_{\text{moves from female to male}} - \underbrace{\left(\frac{f}{f+m}\right)^{\varepsilon}P_{m3}}_{\text{moves from female to female}} \right] \quad (5c)$$

$$\frac{dP_f}{dt} = \underbrace{\left(\frac{\alpha(f+m)}{1+\alpha\beta(f+m)}\right)}_{\text{Total visitation rate}} \left[ \underbrace{\left(\frac{f}{f+m}\right)^{\varepsilon}P_{m3}}_{\text{moves from female to female}} - \underbrace{\left(1-\left(\frac{f}{f+m}\right)^{\varepsilon}\right)P_f}_{\text{moves from female to male}} \right]. \quad (5d)$$

The incorporation of flowering dynamics given in Eq (2) into the system of differential equations for the pollinators model (5) results in a system of ordinary differential equations when $t \leq \min\{\tau_m, \tau_f\}$, before any open flowers begin to close, followed by a system of delay differential equations with a single delay $\tau = \min\{\tau_m, \tau_f\}$ when $\min\{\tau_m, \tau_f\} \leq t \leq \max\{\tau_m, \tau_f\}$, and then by a system of delay differential equations with two fixed delays, $\tau_m$ and $\tau_f$. This model tracks the number of open male and female flowers $(m, f)$ and the number of pollinators of each type $(P_{m1}, P_{m2}, P_{m3}, P_f)$ as they visit male and female flowers.

**Pollination measurement.** The total number of visits to female flowers is an important factor for pollination. Visits to female flowers from the different classes of bees represent different pollen depositions needed to determine success of fruitset.

We define the visits of pollinators to female flowers that result in pollen deposition as either type one, two, or three, as depicted in Fig 2. We then define fruit set for a day $t$ denoted by $P(t)$ as

$$P(t) = 1 - (1-p_1)^{v_1(t)} \times (1-p_2)^{v_2(t)} \times (1-p_3)^{v_3(t)} \quad (6)$$

where $v_n(t)$ for $n = 1, 2, 3$ represents the total number of type $n$ visits that each flower has received at the time of closing (day $t$), and $p_n$ represents the percent chance that a single visit will fully pollinate a flower to set fruit, for each visit type $n$. The total predicted yield is the fruit set for each day multiplied by the number of female flowers closing on that day, summed over all the days,

$$\text{The total predicted yield} = \sum_t DFC(t) * P(t). \quad (7)$$

where $DFC(t)$ denotes the daily number of female flowers closing at day $t$. The total predicted

yield proportion over all days is the number of flowers closing on each day multiplied by the fruit set for that day divided by the total number of female flowers, in our calculation we use the number of total female flower buds,

$$\text{The total predicted yield proportion} = \frac{\sum_t DFC(t) * P(t)}{B_f}. \tag{8}$$

## Parameterization

All model parameters are listed in Table 2. In order to parameterize the visitation rate Eq (3) we assume the pollinators are active in the field for only 4 hours per day. For the search rate $\alpha$ we assume a pollinator encounters 2 flowers per min, or 480 visits per day, assuming they forage only 4 hours a day. For the handling time $\beta$ it has been observed that the average time a pollinator spends on a flower is 16 sec, or 0.0011 days [36]. We use an odds ratio to parameterize the preference parameters, $\delta$ and $\epsilon$.

Experimental observations on pollinator behaviors in environments with equal density of male and female flowers (1:1 ratio, which is typical across planting regimes even when the ratio of male to female plants differs, due to flower pruning practices) reveal that pollinators on male flowers have a 0.957 probability of remaining on male flowers and those on female flowers have a 0.951 probability remaining on female flowers [14, 24, 34]. This information was used to help parameterize the preference parameters. Following the experimental conditions, we assume an equal density of male and female flowers and take $\delta = \ln(0.957)/\ln(0.5)$ and $\epsilon = \ln(0.951)/\ln(0.5)$. It is important to note, that while these preference parameters are constant, the probability of switching from flower types depends on these preferences, as well as the open number of male and female flowers, see Eq 4 and Fig 3.

Once flowers begin to open, the peak day for flower openings occurs between days 2 and 9 [24, 37, 38] and we assume $t_m = 6$ and $t_f = 6$ days. Flowers remain open for 3–7 days [30] and we assume base values of 5 for $\tau_m$ and 4 for $\tau_f$. Observed pollinator densities range from 0.2 – 20 per 1000 flowers [30], and we assume a baseline value of $\rho = 6$ pollinators. We assume the

**Table 2. Model parameters, base values and ranges used in simulations.**

| Parameter | Meaning | Units | Base value | Range | References |
|---|---|---|---|---|---|
| $\alpha$ | search rate | 1/(day×flower) | 480 | 120—3600 | [14] |
| $\beta$ | handling time | days | 0.0011 | 0.00013– 0.0094 | [22, 23, 36, 39] |
| $\delta$ | preference to remain on male flowers | – | 0.0634 | 0–1 | [14, 22, 24, 33, 34] |
| $\varepsilon$ | preference to remain on female flowers | – | 0.0725 | 0 –1 | [14, 22] |
| $B_m$ | number of male buds | flower | 600000 | 300000–900000 | [18, 37] |
| $B_f$ | number of female buds | flower | 600000 | 300000–900000 | [37] |
| $t_m$ | peak day of male flower opening rate | day | 6 | 2—9 | [24, 37, 38, 40] |
| $t_f$ | peak day of female flower opening rate | day | 6 | 2—9 | [24, 37, 38, 40] |
| $\sigma_m$ | spread of male flowering period | | 2.5 | 0.5–5.5 | [24, 37, 38] |
| $\sigma_f$ | spread of female flower period | | 2 | 1–4 | [24, 37, 38] |
| $\tau_m$ | life span of male flowering | day | 4 | 3–5 | [30] |
| $\tau_f$ | life span of female flower | day | 5 | 3–7 | [34, 41, 42] |
| $\rho$ | pollinators per 1000 flowers | pollinators/flower | 6 | 1—20 | [18, 30] |
| $p_1$ | percent chance to set fruit from single type one visit | | 0.66 | 0.25– 0.75 | [14] |
| $p_2$ | percent chance to set fruit from single type two visit | | 0.55 | 0.1– 0.65 | [14] |
| $p_3$ | percent chance to set fruit from single type three visit | | 0.22 | 0– 0.5 | [14] |

percent chance that a single type one visit (transitions a pollinator from group $P_{m1}$ to group $P_{m2}$) will fully pollinate a flower to set fruit is $p_1$ = 66%. A single type two visit (transitions from $P_{m2}$ to $P_{m3}$) will fully pollinate a flower with assumed $p_2$ = 55% and a single type three visit (transitions from $P_{m3}$ to $P_f$) will fully pollinate a flower with assumed $p_2$ = 22% [14]. For the total number of flowers we assume the number of flower buds follows $B_m = B_f$ = 600, 000 per ha.

## Model simulations

All simulations were conducted using Matlab's differential equations solvers ode45 and dde23 with initial conditions such that 0% of pollinators were $P_{m1}$, $P_{m2}$, and $P_{m3}$, and 100% of pollinators were $P_f$ at time $t = 0$ for an orchard of sample size of 1 ha. Parameter values for the total number of flower buds $B_m$ (male) and $B_f$ (female) along with the number of pollinators per 1000 female flowers $\rho$ are used to determine the total number of pollinators for each simulation.

**Base simulations testing model behaviors.** Model simulations for the set of baseline values given in Table 2 are shown in Fig 4. Pollinators of type $P_{m1}$ and $P_f$ fluctuate during the blooming period while the number of pollinators of types $P_{m2}$ and $P_{m3}$ remain low (Fig 4a). The accumulated number of visits to female flowers at the time of closing is almost identical across visit types (Fig 4e), and is driven by the number of open female and male flowers, since the number of pollinators is fixed. Our model output measure (total predicted yield) is shown in Fig 4f. As expected, type one visits have the highest fruit set rate while type three visits have the lowest fruit set rate, even though the total number of these visits are similar (Fig 4f). The results in Fig 4(f) multiplied by the daily number of female flowers closing yields the daily predicted yield. Then the summation of this yield returns the main output for our model; the total predicted yield (see Eq (7)). Under these baselines values total predicted yield is 545,120 fruit / ha with a predicted fruitset of 90%. This is on the high end of reported fruit set in "Hayward" orchards (c.f. 80% in Gonzales et al. 1998 [43]), but matches the experience of the authors in field trials where fruit set is measured before harvest and thus is a higher figure than fruit set calculated for yield (Pattemore D pers. obs., Broussard M pers. obs.). Accordingly, the fruit number per hectare is higher than the 200,000—300,000 often reported in the literature [44–46], but again is within the range of possible outcomes.

**Simulations exploring model outputs.** We varied key model input parameters and investigated model predictions with numerical simulations and sensitivity analysis. Model parameters are presented in Table 2. A major model output measure is the predicted yield, which is defined as the number of female flowers per ha that became fully pollinated fruit. A second important model output is the percentage of female flowers that became fully pollinated fruit, defined here as the fruit set. We used numerical simulations to explore variations in flowering dynamics including the percentage of buds that are female and shifts in the duration of time when male and female flowers are both opened (by varying the peak date in male flower opening rate). We also explored variations in pollinators dynamics including bee densities, preference parameters and pollinator handling time.

**Parameter sensitivity analysis.** In order to better assess the predictions of our model we investigate the uncertainty of the estimated parameter values using Latin Hypercube Sampling (LHS) with the statistical Partial Rank Correlation Coefficient (PRCC) technique, which provides a global parameter sensitivity analysis. LHS is a stratified Monte Carlo sampling method without replacement giving a global and unbiased selection of parameter values [27]. The PRCC technique is used to assess the importance of each parameter for a given output measure. It is appropriate when the parameters have a nonlinear and monotonic relationship with

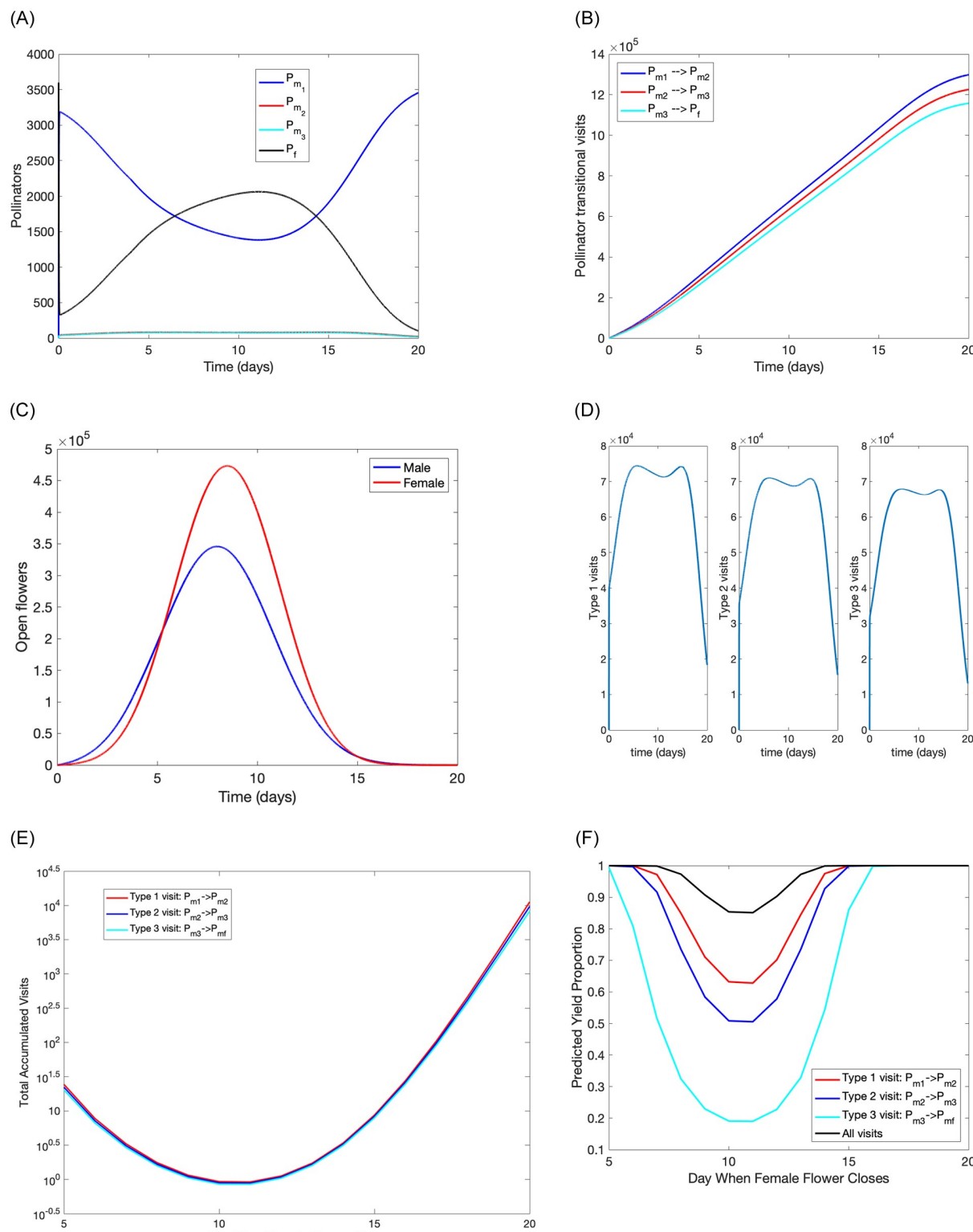

**Fig 4. Model simulations presenting (a) number of pollinators during the bloom period, (b) accumulated number of transitional visits of different types, (c) number of open male and female flower during the bloom period, (d) daily number of transitional visits of different types, (e) daily number of transitional visits of different types per female flower at the time of closing, (f) predicted number of fully pollinated fruit for each type visit and all visits for female flowers at the time of closing.** All parameter values are base values in Table 2 with initial conditions that pollinators haven't collected any pollen yet (*i.e.*, $P_{m1} = P_{m2} = P_{m3} = 0$ and $P_f = \rho^* B_f / 1000$).

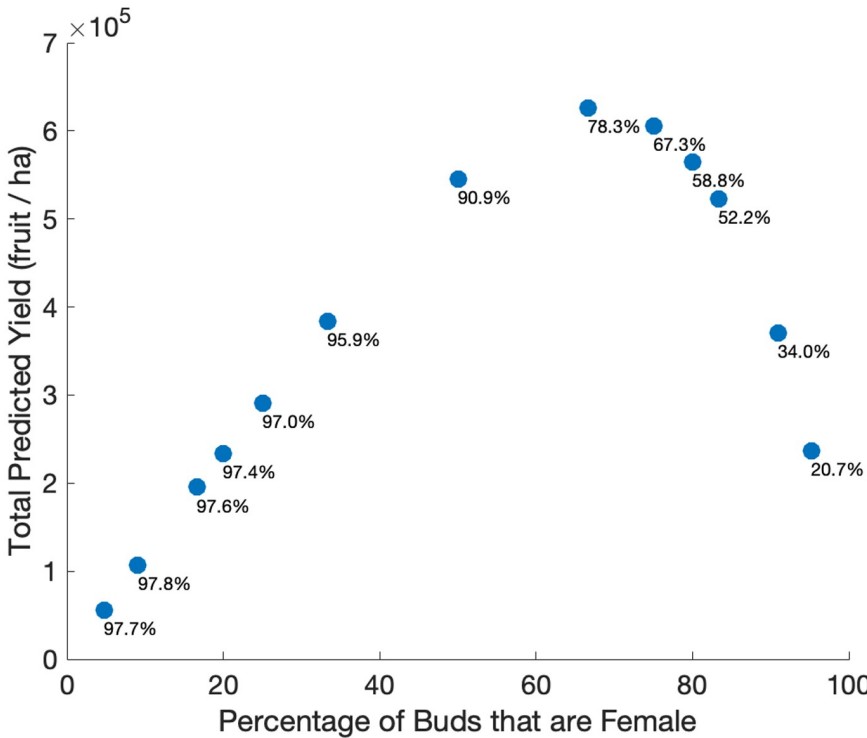

**Fig 5. Total predicted yield (fruit per ha) as a function of the percentage of buds that are female.** The total number of buds was kept constant at 1.2 million/ha and the fraction of female flowers varied. Other parameters are baseline values in Table 2. The fruitset (percentage of open female flowers that achieved full pollination) is listed under each data point.

the output measures. Using a model orchard of 1 ha we used LHS to sample the parameters listed in Table 2 and used PRCC to investigate the output measure of the total predicted fully pollinated fruit per hectare (yield). Following Marino et al. 2008 [27] we performed a z-test on the resulting PRCC values and verified that, in general, higher magnitude PRCC values correspond with a stronger influence on the output measure. Most of the parameters had nonlinear and monotonic relationships to the total predicted yield. Additional investigation on parameter values that were nonmonotonic was done by truncating the parameter space to monotonic regions, details are presented in the appendix.

## Results

To investigate the role of key plant parameters we varied the ratio of male to female flowers in the orchard by fixing the total number of flowers and varying the percentage of flowers that are female, all other parameters were set to their base values shown in Table 2. Increasing the fraction of flowers that are female (versus male) per hectare first increases the total predicted yield (fruit per hectare), peaking near 0.66, and then decreasing rapidly as female flowers make up the majority of the orchard (Fig 5). When the fraction of female flowers per hectare is low, nearly all female flowers produce fruit: predicted fruitset reaches above 97%. However, the total yield (fruit produced) is low due to the low quantity of female flower buds. On the other hand, when most flowers are female, predicted fruitset decreases to 20% along with an associated decline in yield. This is due to the fact that while the quantity of female flowers is high, the quantity of male flowers is low and the chances of successful pollination decreases

substantially. The model predicts a maximum fruit yield when female flowers make up two thirds of the field with a fruitset (percent of open flowers that achieved successful pollination) of 78.3%.

Other key plant parameters influence the timing of when male and female flowers are open and receptive. Pruning and the use of chemical bioregulators are typically used to control the onset and duration of flowering by growers [47]. Over a longer time frame, cultivar selection can be used to ensure adequate overlap of male and female flowering across a range of environmental conditions. The model assumes the rate that flowers open follow normal distributions with key parameters specifying the peak day of flower openings for both the male ($t_m$) and female ($t_f$) distributions. Varying the peak day that male and female flowers open influences the duration of time with both types of flowers open simultaneously as well as the number of flowers open during these times (Fig 6a). In particular, differences between $t_m$ and $t_f$ shifts these distributions and affects the overlapping time when both flower types are open. In Fig 6 we hold $t_f = 6$ days constant and vary the peak day of male flowers opening from $t_m = 3$–$9$ days. Predicted yield is maximized (with associated fruit set rates above 91%) when both flower types open concurrently with the same peak opening day (Fig 6b). While shifting the peaks a day apart does not have a huge influence on the predicted yield, a shift of two or three days has significant consequences.

To investigate the role of key pollinator parameters, we varied pollinator density based on data on observed honey bee densities. The total predicted yield increases rapidly as the number of bees increases from one to six bees per 1000 female flowers (Fig 7). Here fruit set also increases from 39% with only one bee per 1000 female flowers to over 90% with six bees per 1000 female flowers. While continuing to increase the number of bees does increase fruit set rate and the total predicted yield, the increase slows down substantially above six bees per 1000 female flowers.

Pollinator behavior parameters also play important roles in the model. The model includes preference parameters for pollinators to remain on the type of flower they are visiting, based again on data from honey bee observations. For the baseline values, a pollinator on a male flower preferentially chooses to visit a male flower next ($\delta$), likewise a pollinator on a female flower preferentially chooses to visit a female flower next ($\epsilon$). Total predicted yield increases as the pollinators increasingly prefer to switch between male and female flowers in sequential visits (Fig 8a). The yield increases substantially when preference for switching is very small and saturates quickly after. The drastic increase in yield begins to plateau close to the baseline parameter values for the preferences, $\epsilon$ and $\delta$ (dashed lines in Fig 8a). It is important to note that Fig 8a explores variations in preference parameters for a constant male to female flower ratio. The actual probabilities of moving between types of flowers is described in Eq (4) and variations in preference parameters are depicted in Fig 3 across different orchard conditions.

Another relevant pollinator behavior is the speed of foraging. Our model includes two parameters for this: the handling time and search rate. Our analyses indicate that of these two, the handling time is the most influential; the total predicted yield decreases quickly as the pollinators' handling time increases (Fig 8b). When the handling time increases from 10 sec to 60 sec, fruit set rates decrease from 100% to 50%.

The modeling framework enables us to vary key plant and pollinator parameters simultaneously. For a given percentage of female flower buds that make up the orchard, predicted yield increases as the number of bees per 1000 female flowers increases (Fig 9). When the female flower buds percentage is high (between 50% and 90%), maintenance of bee densities over 6 bees per 1000 female flowers will lead to better pollination and therefore ensure a high predicted yield.

(A)

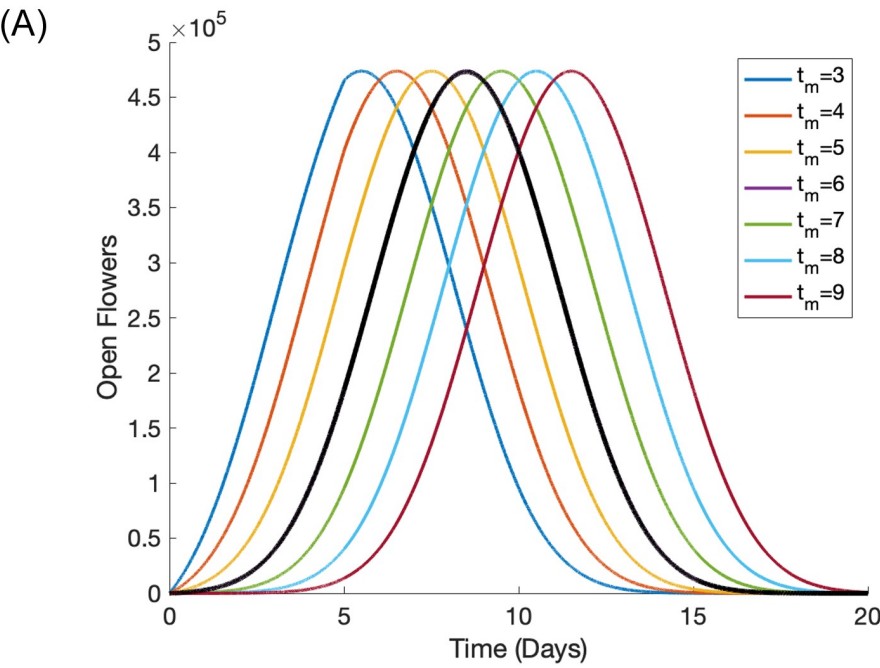

(B)

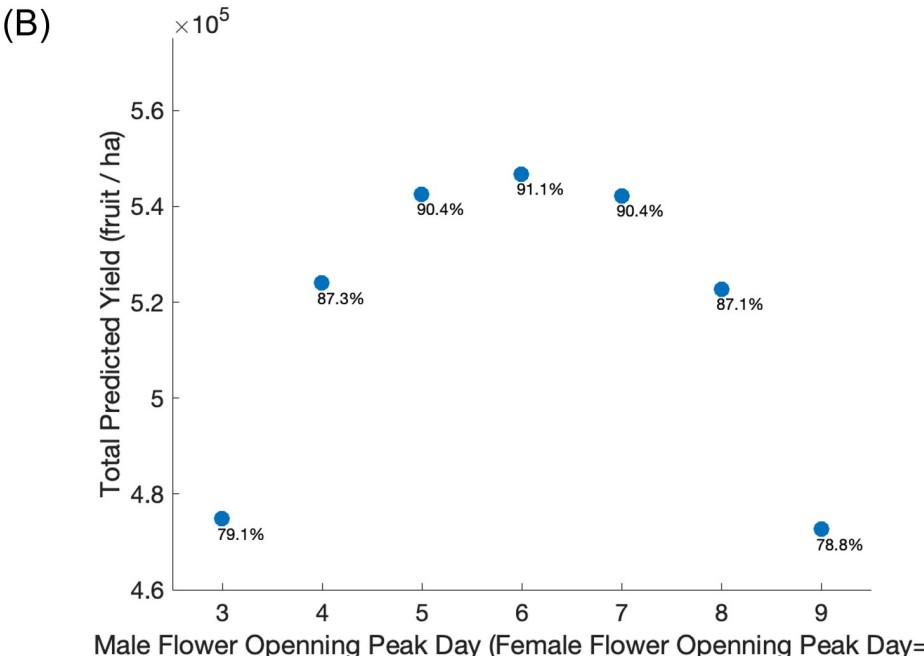

**Fig 6. Open male (colored) and female (black) flowers (a) and total predicted yield (b) for varying peak day of male flower opening from day 3 to day 9.** The total amount of buds was kept constant at 1.2 million/ha with a male to female flower ratio of 1:1 and 6 bees per 1000 female flowers. Other parameters are baseline values in Table 2. The percentage of open female flowers that achieved sufficient pollination to set fruit is listed under each data point.

Parameter sensitivity analysis shows that the percentage of female flowers, the total number of buds, and the bee density have the most significant effect on the total predicted yield (Fig 10) with a positive correlation. Bee density, the pollinators' preference to switch from female to male flowers ($\epsilon$), the male flowering period ($\sigma_m$), and the pollinator's preference to switch

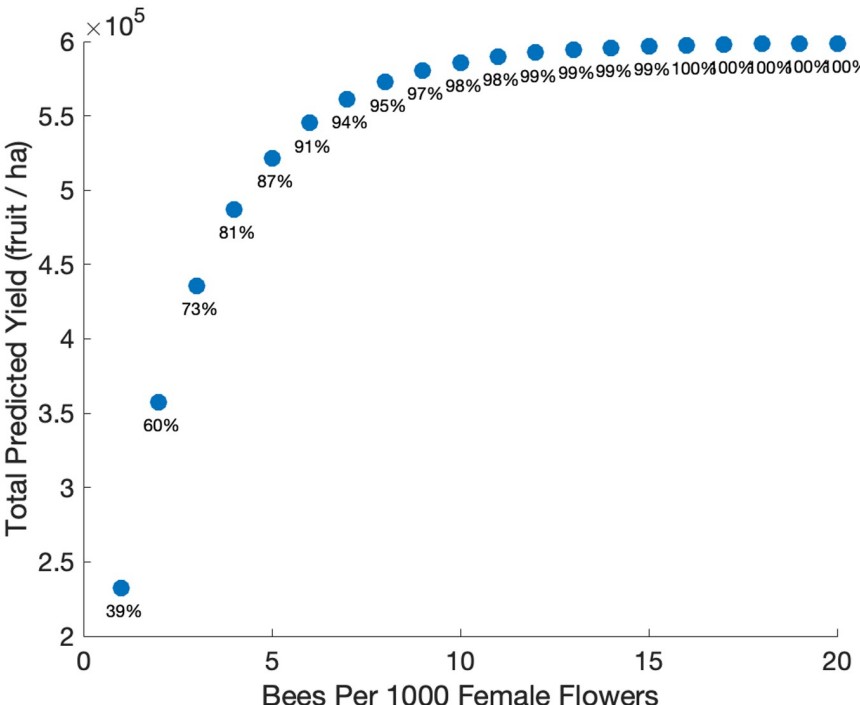

**Fig 7. Total predicted yield as a function of the number of bees per 1000 female flowers.** The bee density varies from 1 bee per 1000 female flowers to 20 bees per 1000 female flowers, and the total amount of buds was kept constant at 1.2 million/ha with a male to female flower ratio of 1:1. Other parameters are baseline values in Table 2. The percentage of open female flowers that achieved sufficient pollination to set fruit is listed under each data point.

from male to female flowers ($\delta$) are the next most important parameters that are positively correlated with the predicted yield, while pollinator handling time is the only parameter with a strongly negative effect on the total predicted yield.

## Discussion

Flower density and the percentage of female flowers were highly influential parameters in predicting final fruit yield. Also important was the width of the male blooming window. Managed honey bees are the primary mode of kiwifruit pollination globally [2], and several pollinator-related factors were found to influence yield, with bee density, flower handling time, and preference for moving between flowers of different sexes all highlighted by our sensitivity analysis.

Kiwifruit flowers may take up to 40 honey bee visits to be fully pollinated [48], but this is partially due to the large numbers of bees which deposit little or no pollen. We found that increasing bee density will increase fruit production, but that there is a point of diminishing returns after the first 6-8 bees per 1000 female flowers and buds. This finding broadly agrees with the literature, which reports that densities of around 3-6 bees per 1000 flowers are sufficient for full pollination [25, 49, 50], with sustained higher bee numbers being unusual, though sustained densities of 14 bees per 1000 flowers have been reported in cages [49] and densities of 30-60 bees per 1000 flowers may occur for a very brief period of time in rare circumstances [4, 34]. We found that a longer flower handling time was negatively correlated with fruit production in this model. Although empirical data show that honey bee flower handling time is not correlated with pollen deposition [14], the rate of flower visitation is a well-known factor in limiting the effectiveness of pollinators independently of pollen deposition. [51].

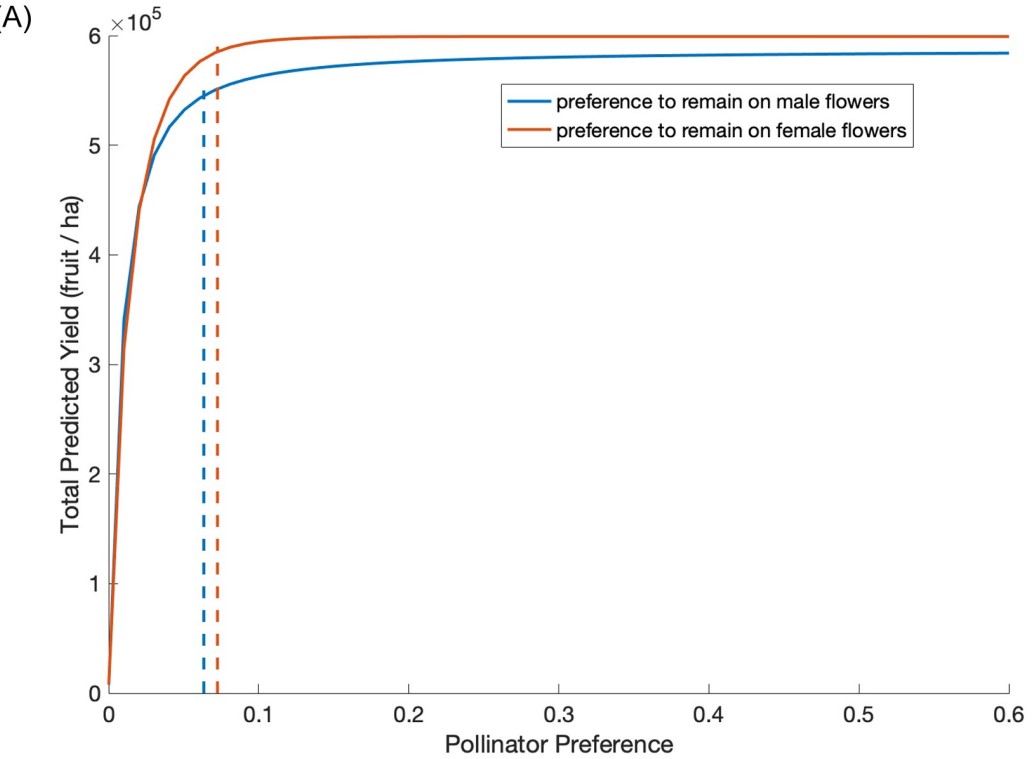

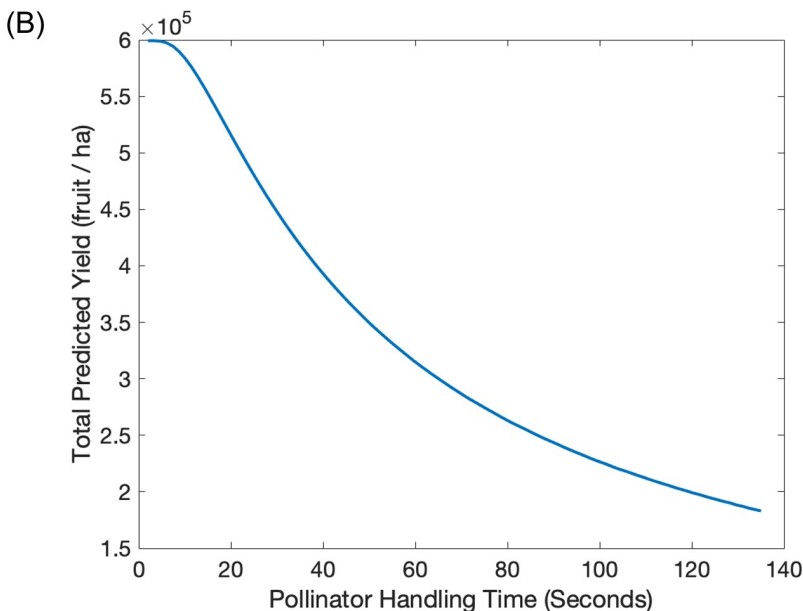

**Fig 8. Total predicted yield for varying pollinators' preference of flowers (a) and for varying pollinators' handling time (b).** Other parameters are baseline values in Table 2. Pollinators prefer flowers of the same sex in sequential visits; in (a) low preference values near 0 correspond with strong tendencies to remain on either male or female flowers, higher preference values correspond with strong tendicies to switch flower type. Blue dashed line in (a) depicts baseline values of $\delta$ (preference to remain on male flowers) and red dashed line depicts values of $\epsilon$ (preference to remain on female flowers).

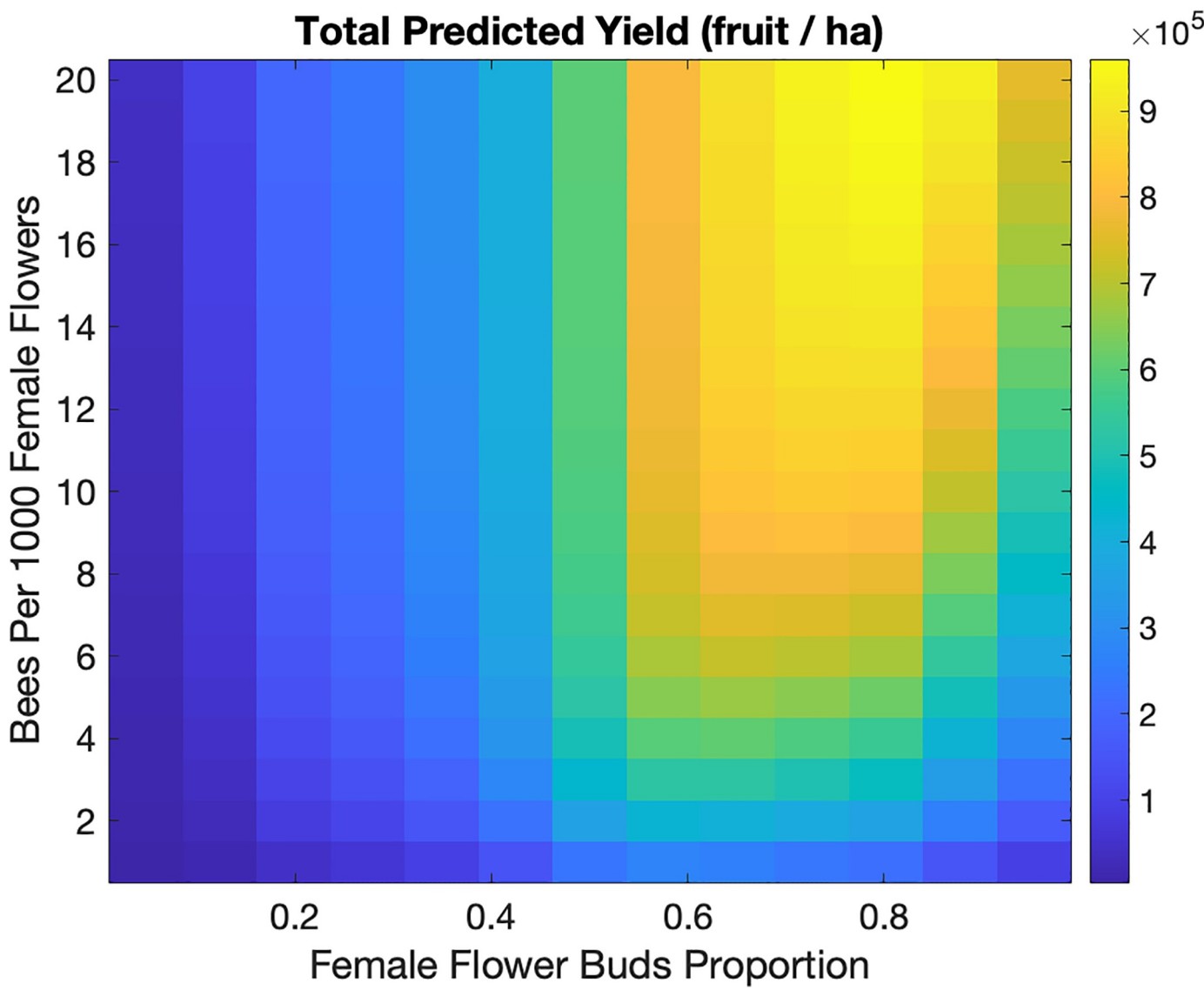

**Fig 9. Total predicted yield for varying the proportion of female flower buds and the number of bees per 1000 female flowers.** The total amount of buds was kept constant at 1.2 million/ha. Other parameters are baseline values in Table 2.

Preference factors are less well-known, but highlighted here. Honey bees are able to differentiate between male and female kiwifruit flowers without landing on them [34], and they must travel from a male flower to a female flower to deposit viable pollen. This chance of switching can potentially be affected by other pollinators in the field [52], as well as the attractiveness of the male and female cultivars. Increasing the chance of switching between plant sexes may be a critical factor for kiwifruit pollination, as the baseline values in our model are right on the edge of a steep decline—if less switching happens than currently reported in the literature (as indicated by the base parameter values), there could be very significant, negative impacts on pollination.

When examining the interaction of bee density and the proportion of female flowers, we found that, at typical bee densities ($< 12$ bees per 1000 flowers), the optimum proportion of

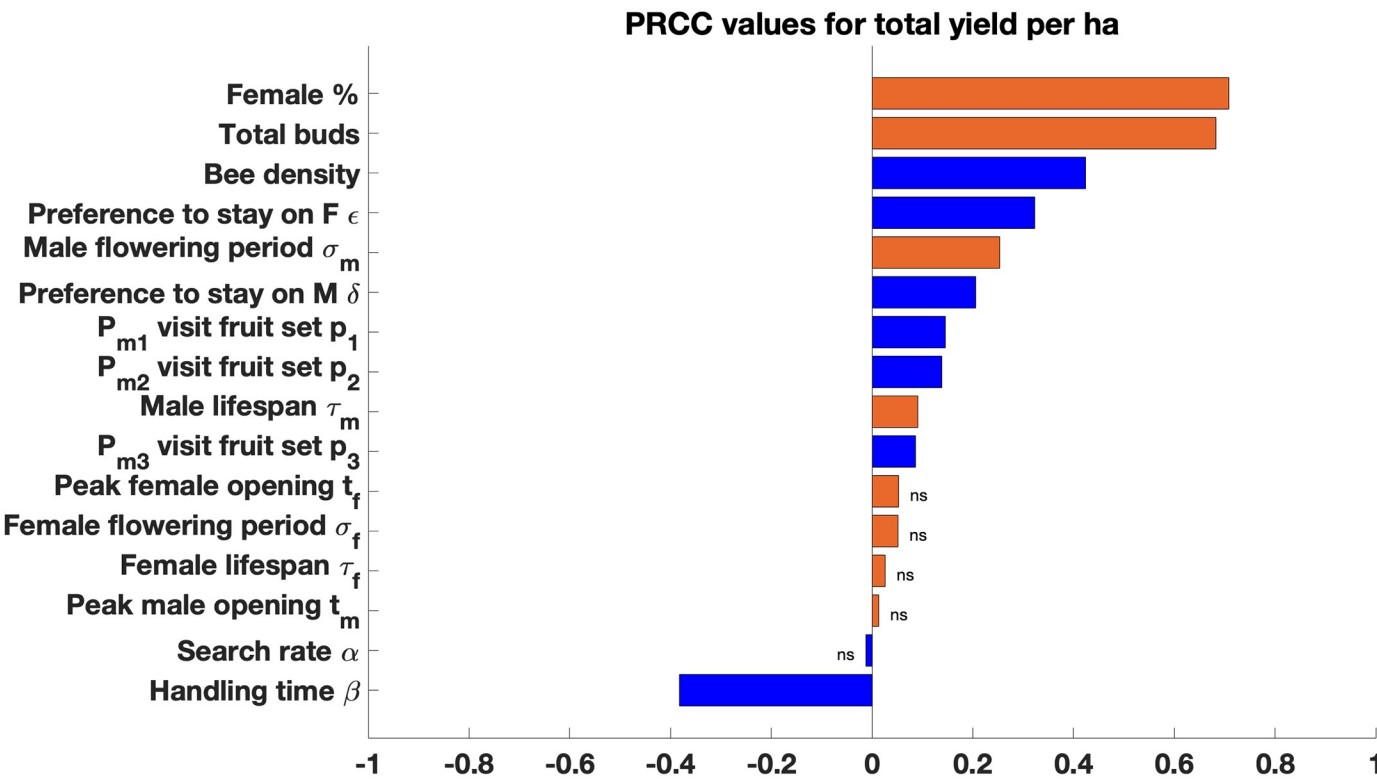

**Fig 10. Sensitivity analysis of the delay differential equation model using partial rank correlation coefficient (PRCC) values for each parameter in the Latin hypercube sampling.** PRCC values marked as ns are not significant ($P \geq 0.05$). Flower traits are in orange and pollinator traits are in blue.

female flowers was 65-75% of total flowers, representing a 'sweet spot' between having more possibilities for fruit development and risk from insufficient movement of bees between the two flower sexes. Current orchard plantings have an approximately 50:50 ratio between male and female flowers [4], highlighting an opportunity to increase yield by changing pruning practices to increase the proportion of female flowers-an easily achievable intervention compared with changing pollinator behavior.

Our model takes advantage of over 30 years of field-based data in New Zealand and other parts of the world and provides a way to quantitatively assess how different plant- and insect-related factors interact and their importance for final fruit set. Our results suggest that choosing cultivars which have their peak bloom on the same day, planting and pruning to achieve approximately 70% female flowers in the orchard, having as many flowers as the vine can support to full fruit size, and placing enough hives to maintain more than 6 bees per 1000 flowers will optimize yield. There is the potential for future work to improve the predictive power of this model by accounting for multiple pollinators and spatial scale and pattern.

## Supporting information

**S1 Appendix. Many of the parameters have monotonic relationships with the output measure (S1a Fig) and the PRCC statistics for those are reliable.** However, we note that parameters $\sigma_m$, $\sigma_f$, $t_m$, $t_f$ and the proportion of female flower buds in the field exhibit nonmonotic behaviors. Therefore, we conducted additional LHE sampling by truncating the ranges of these parameters to monotonic regions. S1b and S1c Fig depict the monotonicity of the

truncated parameter space. The resulting PRCC results for the entire parameter space as well as the truncated parameter spaces are compared in S2 Fig. Parameters for the total number of buds, percentage of female buds, bee density, and handling time are consistently identified as important parameters in all cases. We note that in the truncated case we split the percentage of female flower buds into the cases of 5–76% and 76–96%. In the first half this parameter shows a highly influential positive relationship with predicted yield (large positive PRCC value) and in the second half the parameter is inversely related to the predicted yield. This is as expected as saturating the field with only female buds will eventually cause a decrease in yield. These dynamics are observed in the monotonicity plots as well.
(PDF)

**S1 Available code. Matlab code for our model was provided as an online supplemental file and is available to download.**
(M)

## Acknowledgments

We would like to thank Mark Goodwin for his assistance in obtaining data for model parameterization, and Ruth Williams and Warrick Nelson for their feedback on the manuscript.

## Author Contributions

**Conceptualization:** Angela Peace, David Pattemore, Melissa Broussard, Dilini Fonseka, Nathan Tomer, Nilsa A. Bosque-Pérez, David Crowder, Allison K. Shaw, Linley Jesson, Brad G. Howlett, Mateusz Jochym, Jing Li.

**Formal analysis:** Angela Peace, Melissa Broussard, Dilini Fonseka, Nathan Tomer, Mateusz Jochym, Jing Li.

**Supervision:** David Pattemore.

**Writing – original draft:** Angela Peace, David Pattemore, Melissa Broussard, Dilini Fonseka, Nathan Tomer, Nilsa A. Bosque-Pérez, David Crowder, Allison K. Shaw, Linley Jesson, Brad G. Howlett, Mateusz Jochym, Jing Li.

**Writing – review & editing:** Angela Peace, David Pattemore, Melissa Broussard, Dilini Fonseka, Nathan Tomer, Nilsa A. Bosque-Pérez, David Crowder, Allison K. Shaw, Linley Jesson, Brad G. Howlett, Mateusz Jochym, Jing Li.

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
