## [Decision Letter · Decision Letter 0]

8 Jun 2020

PONE-D-20-07351

Orchard layout and plant traits influence fruit yield more strongly than pollinator behaviour and density in a dioecious crop

PLOS ONE

Dear Dr. Peace,

Thank you for submitting your manuscript to PLOS ONE. After careful consideration, we feel that it has merit but does not fully meet PLOS ONE’s publication criteria as it currently stands. Therefore, we invite you to submit a revised version of the manuscript that addresses the points raised during the review process.

I have been able to source two reviews from researchers that have expertise on both modelling and plant pollination systems. Both reviewers see the value in your work, as do I, but also made several suggestions that will significantly improve the clarity of the study for the readership of PLOS ONE. Can you please pay careful attention to all suggestions, and especially work to make the definitions etc much clearer (and more consistent). There were also several points in the manuscript where it was not totally clear where parameter data for models had come from. Please be very careful about defining source(s) all empirical data and how this can be accessed from public domain. This enales other researchers to replicate and extend your research. Data can for example be place within table(s) in manuscript; or use one of the public databases if large data sets are involed. If you had a goal that a good HDR could look up your paper and replicate findings, that is a very good outcome and would help ensure others will use this potentially important finding.

We look forward to receiving your revised manuscript.

Kind regards,

Adrian G Dyer, Ph.D.

Academic Editor

PLOS ONE

Journal Requirements:

Reviewers' comments:

Reviewer's Responses to Questions

**Comments to the Author**

1. Is the manuscript technically sound, and do the data support the conclusions?

Reviewer #1: Partly

Reviewer #2: Yes

2. Has the statistical analysis been performed appropriately and rigorously? 

Reviewer #1: Yes

Reviewer #2: Yes

3. Have the authors made all data underlying the findings in their manuscript fully available?

Reviewer #1: No

Reviewer #2: No

4. Is the manuscript presented in an intelligible fashion and written in standard English?

Reviewer #1: Yes

Reviewer #2: Yes

5. Review Comments to the Author

Reviewer #1: This article presents a model of the honeybee pollination of kiwifruit in order to assess how male/female ratio and honeybee stocking rates impact fruit production in an orchard. The research model is described, results are presented and analysed. The problem is clearly defined, an appropriate model is used, and the potential impact of the results is explained. Overall, this paper addresses an important problem using a relevant and appropriate method.

I think a number of issues still need to be addressed prior to considering whether or not this paper should be accepted for publication. If they can be addressed, then I do think the paper warrants publication and I also believe PLoS ONE is a suitable venue.

My main overarching suggestions are:

* that the standard orchard planting ratios be explored and clearly explained to the reader who is unlikely to be familiar with them. These ought to be supported with recent documentation or info. elicited from farmers/growers about current best practice.

* that the paper be restructured so that the reader finds out, as they read the paper, the info. they need to understand what was done and why it was done using the methods/approaches documented. Currently much info. that is needed to understand the first parts of the paper only appears in the later parts of the paper. Specific details are provided below.

* that the paper include a careful explanation and justification for the assumption of male/female ratios used in parameterisation

* that the paper include a careful explanation and justification of how a bee's flower preference could possibly be independent of flower m/f ratios for an insect foraging in the field.

* please label graph axes with units (see below)

* please comment on the utility of the finding about the impact of male opening times to growers. Is it feasible for them to manipulate this trait? Discuss.

* that the model code be placed online if possible and in accordance with any journal policies if required (e.g. GitHub?)

Some detailed remarks:

Line 7: note, not all animal pollinated plants have *insect* pollinators.

Line 10: please cite reference for stocking rates 3-8 col./ha.

Line 17: Explain briefly how this approach could be used to improve conservation practices, e.g. in a sentence or so. Or, drop this sentence altogether.

Line 26... for the non-kiwifruit-experts among your readership (probably most of us readers!) please provide some detail on orchard layouts, standard planting arrangement, physical characteristics of plants including their extent, leaf cover, number of flowers, pruning regimes etc. Otherwise all this is very hard to get a handle on.

Line 34: please, if honeybees are indeed the common managed pollinator of this crop, please provide references to support this (and therefore the choice of honeybees for your model - you only get to this at line 250 currently). Also comment briefly, do bumblebees help? What other pollinators might be playing a role? I see at line 270 you note that the presence of other pollinators may impact honeybee behaviours. For the reader to assess the likelihood of this calling into question the reliability of your results, can you provide your thoughts? Please discuss this in a little detail in the Discussion section also.

Line 50: At this point it looks like an essential aspect of your model assumptions is an unlimited supply of pollen on the male flowers. Please comment explicitly.

lines 53-63: Please can you make this explanation clearer? E.g. simply saying that a move up from f-3-2-1 occurs by visiting male flowers, and a move down occurs from 1-2-3-f by visiting female flowers, and then referring the reader to the figure would be much simpler.

line 68: Please clarify - honeybees are excellent generalists exploiting a wide range of plant species! Surely they are the exemplar polylectic insect (so not olig.!). Doesn't this mean you need to be more careful about putting the case for your selection of Holling-II? E.g., is the fact they are in an orchard which is mono-culture relevant here since they may have no choice of floral resource? Or is there some other reason your assumption works? Please cite references backing up your claim of appropriateness.

EQNs 2a-d. I'm unsure at this point why the equations take the form of the fractions raised to a power. How did you derive this form? (Maybe it is something obvious, sorry if I missed it)

Line 88: You speak here of flower opening rate. Do you perhaps mean that these times tm and tf are times at which the peak *numbers* of flowers are open? Or is this truly the time at which the *rate* of flowers undergoing the change from open to closed is occurring? (I'm just unsure of your intended meaning in this sentence)

Fig 2. refers to a 5x5m orchard. This was initially very confusing to me as I couldn't see why you would model a space so tiny. Only later (cf. line 145) did I realise this fig. caption is the only place 5x5m is mentioned. Can you not instead simply plot this data for the orchard size (1 ha.?) that you actually used for your models? This will save confusing your readers.

Line 104: only below on line 111 does it become clear to me what you mean by this sentence. Please rephrase it to be clear that there is *not* a separate probability parameter or distribution for "fruit set" that is somehow independent of the number of visits. The way 104 is currently worded it seems like this is what you have implemented.

Table 1: Can you comment on the fact that the data you have collected in this table spans a variety of scenarios, even decades and (of course) studies? What does this mean for your study that you have drawn across such a wide variety of conditions to extract parameters for modelling a single orchard?

Lines 128-131: This section is very confusing to me. Surely the chance of a M->F or F->M swap is proportional to the number of flowers of each type in visual or short-term exploratory range, *as well as* the preference of the bee? Treating the preference of the bee as the all important factor and treating the problem of availability as independent of this seems to me to be counter to the likely behaviours of foragers in the field (If the bees were in a y-maze and asked to make a decision about M/F I would expect different results.) This issue re-appears at line 184 and I note you are concerned about its importance at line 270. It really seems to me that this is a key parameter needing careful thought and study.

[[ Aside: Since neither flower offers nectar, may I ask, what is known about why the bees "want" to visit the female flowers at all? If they *could* tell male from female flower 9I see your note on line 267 that this is possible), may I ask, would they not choose male flowers every time so as to forage pollen? (I'm not familiar with the specifics of bee foraging on this particular crop, I guess many readers won't be either, so please state/reiterate in your paper what you (and the literature) know about why these switching probabilities are as they are.) ]]

Line 130: You derive some values for preferences based on an equal number of male/female flowers. As a reader I don't know if this is a sensible assumption. My quick and naive googling online reveals: "The proportion of non-fruit bearing, and thus “unreproductive” male plants in the orchard is generally limited, e.g. often 1 male plant to 5-8 females is advised. This limits availability of pollen." (here: https://www.biobestgroup.com/en/news/the-latest-in-kiwi-pollination). So now I am left wondering about your derivation. Can you explain this so that the reader can follow why you assumed equal numbers of flowers to compute your parameters? You might also wish to explain why then it is okay for you to vary the M:F ratio in your experiments, without recomputing these parameters (see note above on lines 128-131)? [I note your comment on line 281 that 50:50 ratio is normal... is this based on current practice? The paper you cite is 2012. Is there a newer reference? (Or, if not, is there evidence that nothing has changed from talking to a commercial farmer might help support your claim?)]

Line 140: Please can you confirm, and cite ref's supporting the idea, that the male and female flowers are both open for the same length of time? Is it true that the pollen from a particular male flower is viable for the same length of time that the stigma of the female flowers are receptive? (This isn't necessarily the case). If not, does your model need to take this into account somehow?

Line 155: You write, "Intuitively... " but actually, haven't you hard-coded 66%, 55% and 22% as the chance that a visit of type 1,2 or 3 will fully pollinate a flower? So this isn't "intuitive", your code is explicitly written to operate this way. Is that correct? Please can you clarify and rephrase if necessary?

I am not an expert on the Parameter Sensitivity analysis approach so I am not able to comment on how well this has been conducted.

Lines 201-202. It wasn't clear from the text above that this variation was going to be explored. Please can you state above that this was something you would investigate? (It seems like a good idea). Can you elaborate, is there actually any means by which a grower might manipulate flowering time of kiwifruit? I.e. could a grower capitalise on this? How? Or is it beyond their control?

Fig 3d) Please match vertical axis scales in all three parts of the figure.

3f) caption... spelling of pollinated.

Fig 4. Please use a numerical value scale that matches the unit (i.e. % not decimals).

Fig. 6 - vertical axis unit = fruit per ha.? Please write it on the graph.

Fig. 7. Caption: Blue dashed line [singular] in (a) depict*s*... and and red dashed lines [which line*s? there's only one line [singular] depict*s*...

Fig. 7(a) Graph: I struggle to read this. Firstly, please place a unit (fruit count?) on the axis. Secondly, aren't the yields per ha. going to be for a specific *pair* of M->F AND F->M values? How do I read a yield for a specific pair of these values off the graph? I can only see how yield corresponds to a single value of EITHER M->F OR F->M switching preference. Or have I really misunderstood what is being shown in this figure? If so, please clarify for me.

Fig. 9 caption (P???) - scrambled text on my PDF copy

Please check references 16, 24, 38 - they are scrambled or incomplete.

Reviewer #2: Recommendation: this article has sufficient potential, with minor revisions.

The authors have used historical empirical data to parameterise their mathematical modelling (delay differential equations with Latin hypercube sampling and sensitivity analysis) to test the relative importance of pollinator behaviour and plant biology, both singly and simultaneously, on fruit yield in kiwi fruits grown in New Zealand. While I understand mathematical modelling is important for simulating the often-complex processes observed in biological systems which can enable effective generation of hypothesis, modelling is often only as good as the data used to create the model. This makes the paper hard to assess the authors claims without critically reviewing the source material used in this model. I would recommend a summary of the empirical data used and extracted in these models. However, I believe the mathematics behind the modelling appears to be sound and nicely incorporates, plant and pollinator data and has the potential to inform kiwifruit growers how to improve yields based on plant ratio/pruning/layout and stocking rate of honeybee pollinators.

The model predicted the following based on such data (from the abstract):

1. At realistic bee densities, the optimal orchard had 65-75% female flowers

2. The most benefit was gained from the first 6-8 bees/1000 flowers, with diminishing returns thereafter

3. Bee density significantly impacted fruit production

4. Plant-based parameters of flower density and male-to-female flower ratio were the most influential

My main concerns about the paper is the decisions and data behind the assumptions and base-line parameters used in the modelling. Generally, I believe the title, abstract and introduction needs to be re-worked so that they are less confusing and flows better. The materials and methods, results and discussion sections are more logical and flow reasonably well. References need to be checked again (some references are incomplete).

Title

The title makes no mention of the fact that results were obtained by mathematical modelling.

The title does not inform the reader of what type of crop was used (Actinidia deliciosa ‘Hayward’ or A. chinensis?)

The title does not inform the reader what type of pollinator was used (Apis mellifera? Bombus sp.?)

Abstract

Throughout the paper the authors have used interchangeable wording for the variables and parameters used in this model, making it quite confusing. Ie. Pollinator behaviour / insect behaviour / pollinator abundance / stocking rates / pollen loads / flower handling / flower preference / switch preference and ratio of male to female flowers / densities of male and female flower, percentage of female flowers, plant varieties, plant traits, plant biology, flower phenology, yield, orchard layout etc. In order for this to be less confusing, it may be beneficial to list the exact variables and parameters used, their definition and why you used them. I believe more outcomes from the results section could be incorporated and aspects of the last paragraph of the discussion should be used in the abstract as it is written nicely and provides valuable information for the reader.

Introduction

I think the paper would benefit from additional background information about kiwifruit orchards, varieties, pollinators used, yields etc. What is the current industry standard in New Zealand? Please state clearly and uniformly throughout the text which variables and parameters are being incorporated into the model. Hypothesis / predictions could be expanded and written more clearly.

Materials and methods:

I think it would be beneficial to list all assumptions and why you made assumptions.

Figure 1 description – I understand the explanation is in the main text of the materials and methods section, and this may work for some, however it is not obvious from the description what is going on. At least outline the definitions of Pm1, Pm2, Pm3, Pm4.

Figure 2. ‘Example simulations of open flowers over time following equations (5) starting with 1500 of each male and female flower buds Bm = Bf = 1500 (modeling 5 meters By 5 meters orchard field) for (a) m = 1, f = 1, tm = 7, tf = 8, m = 5, f = 6 and

(b) m = 2, f = 4, tm = 7, tf = 8, m = 5, f = 6 and

(c) m = 3, f = 3, tm = 4, tf = 5, m = 4, f = 5’

• What is the assumed biological significance of each example?

What unit for yields?

Figures 5 – 5a and 5b – location of the a) and b) descriptions are confusing.

Figure 7 – 7a and 7b – as above

Results and Discussion

Are written in a clear and logical progression with good discussion.

6. PLOS authors have the option to publish the peer review history of their article (what does this mean?). If published, this will include your full peer review and any attached files.

Reviewer #1: No

Reviewer #2: No

---

## [Author Response · Author response to Decision Letter 0]

13 Aug 2020

Dear Dr. Adrian G Dyer, 

Thank you for your careful review of our manuscript and the invitation to submit a revised version. The two reviewers are clearly knowledgeable and provided helpful feedback that has improved the manuscript. Please see the response to reviewers document for detailed responses to each remark. 

One of the reviewers requested that our matlab code be made available online. We have a model of differential equations and we use matlab's built in solvers to solve it, but we're happy to provide this code if needed. What are your policies for this? Does the journal maintain an online repository? Or can simply add this code as online supplemental material? Thank you.

---

## [Decision Letter · Decision Letter 1]

8 Sep 2020

PONE-D-20-07351R1

Orchard layout and plant traits influence fruit yield more strongly than pollinator behaviour and density in a dioecious crop

PLOS ONE

Dear Dr. Peace,

Thank you for submitting your manuscript to PLOS ONE. After careful consideration, we feel that it has merit but does not fully meet PLOS ONE’s publication criteria as it currently stands. Therefore, we invite you to submit a revised version of the manuscript that addresses the points raised during the review process.

I have had the revised manuscript re-reviewed by reviewer one, who had requested the major revision. Reviewer 1 has found the paper much improved and has requested only a few minor clarifications. If you can make these changes I will accept the manuscript. Thank you for your time to prepare the previous detailed revisions to both reviewers from the previous round of reviewing..

We look forward to receiving your revised manuscript.

Kind regards,

Adrian G Dyer, Ph.D.

Academic Editor

PLOS ONE

Reviewers' comments:

Reviewer's Responses to Questions

**Comments to the Author**

1. If the authors have adequately addressed your comments raised in a previous round of review and you feel that this manuscript is now acceptable for publication, you may indicate that here to bypass the “Comments to the Author” section, enter your conflict of interest statement in the “Confidential to Editor” section, and submit your "Accept" recommendation.

Reviewer #1: (No Response)

2. Is the manuscript technically sound, and do the data support the conclusions?

Reviewer #1: Yes

3. Has the statistical analysis been performed appropriately and rigorously? 

Reviewer #1: Yes

4. Have the authors made all data underlying the findings in their manuscript fully available?

Reviewer #1: Yes

5. Is the manuscript presented in an intelligible fashion and written in standard English?

Reviewer #1: Yes

6. Review Comments to the Author

Reviewer #1: This article presents a model of the honeybee pollination of kiwifruit in order to assess how male/female ratio and honeybee stocking rates impact fruit production in an orchard. The research model is described, results are presented and analysed. The problem is clearly defined, an appropriate model is used, and the potential impact of the results is explained. Overall, this paper addresses an important problem using a relevant and appropriate method.

The paper is much improved after its initial revision. I think the paper warrants publication in PLOS1 with only a few minor revisions to tidy things up. These are detailed below.

My main request is:

Please add a section (or sub-section) on how the model's behaviour was tested... not sensitivity analysis, but how did you do basic checking to ensure it was all functioning as you would expect? E.g. what expected and real-world patterns were you looking to assess against? How did you validate the model's relationship to reality? How did you validate the model's behaviour with respect to your model design?

Some detailed remarks:

ABSTRACT text: "honeybee density per flower" ... "density" seems (to my mind) to conflate two things... i.e. bees / unit-area and bees/flower. Maybe just call it bees/flower as that is how you built the model.

Typos below... (please find these using a word search, there are two copies of the paper in the PDF and if I cite line numbers I am bound to be confusing)

>> in *an* orchard

>> femaler !?

>> Here, chose and varied some parameter*'*s values

>> the text, "to partially compensate for this short-coming of the model we limit active foraging to ..." isn't a clear explanation about the pollen issue. BUT the explanation you gave in the document to the reviewers directly in your response document *is* clear... please can you explain the statement in the paper as you explained it in the response to reviewers' document?

>> under equation (3): "which has the units of *flowers* per *unit of* time" ...perhaps?

>> under equation (4d): "are on verses the proportion *of* female flowers in the..."

>> text above equation (6): "type one visit, the visit that results in transitioning a pollinator from group Pm2 to Pm3 139 as a type two visit, and the visit that results in transitioning a pollinator from group Pm3 to Pf as a type three visit".... you can label these on Fig. 2 and refer the reader to the figure for clarification. You might also consider explaining these "types" with Fig. 2, and referring the reader back to it (instead of explaining the types in this section).

>> the finding of/around Fig 6 is a little obvious... isn't it? If not, please say why this isn't just trivial (i.e. less male flowers open -> less pollen available -> less fruit set).

Please check references 14, 15, 17 - they are incomplete / truncated.

7. PLOS authors have the option to publish the peer review history of their article (what does this mean?). If published, this will include your full peer review and any attached files.

Reviewer #1: No

---

## [Author Response · Author response to Decision Letter 1]

5 Oct 2020

See the uploaded Response to Reviewers document.

---

## [Editor Report · Decision Letter 2]

8 Oct 2020

Orchard layout and plant traits influence fruit yield more strongly than pollinator behaviour and density in a dioecious crop

PONE-D-20-07351R2

Dear Dr. Peace,

We’re pleased to inform you that your manuscript has been judged scientifically suitable for publication and will be formally accepted for publication once it meets all outstanding technical requirements.

Kind regards,

Adrian G Dyer, Ph.D.

Academic Editor

PLOS ONE
---

## [Editor Report · Acceptance letter]

15 Oct 2020

PONE-D-20-07351R2 

Orchard layout and plant traits influence fruit yield more strongly than pollinator behaviour and density in a dioecious crop 

Dear Dr. Peace:

I'm pleased to inform you that your manuscript has been deemed suitable for publication in PLOS ONE. Congratulations! Your manuscript is now with our production department. 

Kind regards, 

on behalf of

Dr. Adrian G Dyer 

Academic Editor

PLOS ONE